# The Effects of Different Natural Plant Extracts on the Formation of Polycyclic Aromatic Hydrocarbons (PAHs) in Roast Duck

**DOI:** 10.3390/foods11142104

**Published:** 2022-07-15

**Authors:** Xixi Shen, Xinyuan Huang, Xiaoyan Tang, Junliang Zhan, Suke Liu

**Affiliations:** 1Key Laboratory of Agro-Product Quality and Safety, Institute of Quality Standard & Testing Technology for Agro-Products, Chinese Academy of Agricultural Sciences, Beijing 100081, China; t2020093@njau.edu.cn (X.S.); 2021808080@stu.njau.edu.cn (X.H.); zhanjunliang1170@163.com (J.Z.); lskdrlwg@163.com (S.L.); 2College of Food Science and Technology, Nanjing Agricultural University, Nanjing 210095, China

**Keywords:** roast duck, polycyclic aromatic hydrocarbons (PAHs), natural extract, antioxidant capacity, inhibition

## Abstract

Polycyclic aromatic hydrocarbons (PAHs) with high carcinogenicity and mutagenicity may be generated in roast duck during high-temperature roasting. Natural extracts with antioxidant effects may inhibit the formation of PAHs. The objective of this study was to compare the effects of green tea extract (GTE); extract of bamboo leaves (EBL); grape seed extract (GSE) and rosemary extract (RE) on PAHs in roast duck to obtain the optimum extract and present a guidance for reducing PAHs in roast duck. The total phenol content and antioxidant capacity of the four extracts were measured, and the PAH changes in the roast duck caused by the four extracts were detected. The total phenol content of GTE was the highest, 277 mg gallic acid equivalent (GAE)/g, while RE was the lowest at 85 mg GAE/g. The antioxidant capacity of RE was 1.9 mmol Trolox/g, which was significantly lower than that of the other three. The four extracts inhibited PAHs formation in roast duck to varying degrees: When the concentration was 25 g/kg, the best inhibitory effects on Benzo [a] pyrene (BaP) and PAH4 (BaP, BaA, BbF and CHR) were obtained from GTE, with inhibition rates of 75.8% and 79.7%, respectively, while the weakest inhibition rates, 32.7% and 43.6%, respectively, were from RE.

## 1. Introduction

Polycyclic aromatic hydrocarbons (PAHs) are a class of hydrocarbons linked by two or more benzene rings through a fused ring or a single bond. They form mainly due to incomplete combustion or the pyrolysis of organic components, including fat, protein and carbohydrates at temperatures over 200 °C [1]. Most PAHs are teratogenic, carcinogenic and mutagenic to humans [2,3,4]. According to their carcinogenicity and mutagenicity, 16 PAHs are currently listed in the Priority Control Pollutant List by the International Agency for Research on Cancer (IARC) and the European Union Environmental Protection Agency. Four polycyclic aromatic hydrocarbons (PAH4), benzo [a] pyrene (BaP), benz [a] anthracene (BaA), benzo [b] fluoranthene (BbF) and chrysene (CHR), are suitable indicators for PAH occurrence and toxicity in food. Among the PAH4, BaP was classified into group 1 (carcinogenic to humans), and the three others were classified into group 2B (possibly carcinogenic to humans) by the IARC [5]. Several studies have been carried out to investigate its presence in foods and feeds [5,6]. Data from the European Food Safety Authority (EFSA) indicated that the main source of exposure to PAHs in non-smokers was certain foods [7].

In recent years, various studies have found that PAHs in foods were mainly produced by high-temperature pyrolysis, gas combustion or direct contact with open fire (such as baking, grilling [8], drying [9] and smoking [10]), mainly due to the incomplete combustion of organic matter [11,12,13] or the high-temperature degradation of fat in barbecue products [14].

As a roasted meat product, Beijing roast duck is a famous traditional Chinese dish with a long history and wins its popularity among tourists all over the world. However, its unique cooking method, high subcutaneous fat content and added condiments are more likely to produce carcinogens including PAHs at high temperature, which has also drawn extensive concerns.

The main mechanism of PAH formation is the recombination of free radicals produced during the combustion of food molecules under harsh conditions; these products eventually build up in food items [15].

Antioxidants have proved to interfere with these radical reactions or trap free radicals, for instance, intermediates of heterocyclic aromatic amines (HCAs) and PAHs, to prevent HCA and PAH formation [16,17]. Therefore, adding components of higher antioxidant activity and scavenging and inhibiting the production of free radicals are effective means of controlling the formation of PAHs. The inhibitory effect of some natural extracts on the formation of PAHs in meat products has been documented [18,19,20]. Studies have shown that marinating meat with spices such as beer [21], garlic and onions [22] can remove the free radicals produced during high-temperature cooking, reducing the levels of PAHs in the final product [21]. In addition, during the barbecue process of meat products, the temperature and time should be reasonably controlled; the barbecue mode was changed, and raw meat with low fat content effectively reduced the content of PAHs in barbecue meat products [23,24]. Fei Lu et al. [25] compared the formation of PAHs in beef and chicken meatballs fried at 180 °C by adding garlic, onion, red pepper, red pepper powder, ginger and black pepper to evaluate the inhibitory effects of spices on PAHs in the meatballs. The results showed that the antioxidant capacity of the spices determined the efficiency of inhibiting the formation of HCAs and PAHs. Studies had shown that green tea extract (GTE) [20,26,27], extract of bamboo leaves (EBL) [28,29,30], grape seed extract (GSE) [18,31] and rosemary extract (RE) [32,33,34]) have antioxidant capacity.

This study selected GTE, EBL, GSE and RE at different concentrations to add to the pickling solution of roast duck to investigate the inhibitory effect of different extracts on the formation of polycyclic aromatic hydrocarbons in roast duck.

## 2. Material and Method

### 2.1. Material

Raw duck (2.00 ± 0.30 kg) was purchased from Wuyi Jianong Agricultural Products Co., Ltd.(Hengshui, China).

Rosemary extract (RE, purity 10%), grape seed extract (GSE, 95% OPC proanthocyanidins, purity 10%), green tea extract (GTE, purity 10%) and extract of bamboo leaves (EBL, purity 10%) were obtained from Xi’an Ruitian Biotechnology Co., Ltd. (Xi’an, China). The four extracts are water-soluble powders. Extract of 30 mg was accurately weighed, dissolved in 8 mL ultrapure water and extracted by ultrasonics for 10 min, and then the supernatant was filtered for later use.

Fennel, cinnamon, pepper, clove, fragrant leaves, cardamom, nutmeg, ginger, green onion, soy sauce, garlic, salt, sugar, monosodium glutamate, cooking wine and water were mixed, and the natural extract powders were added at three levels, 5, 15 and 25 g/kg, to prepare pickling solutions for roast duck.

### 2.2. Design of the Experiment

In this experiment, four natural plant extracts with antioxidant activity were added to the pickling solution: grape seed extract (GSE), rosemary extract (RE), bamboo leaf extract (EBL) and green tea extract (GTE) to evaluate their effects on the production of 16 kinds of PAHs in the skin of roast duck under roasting conditions. The main factors were the type and amount of the extracts. Each extract was set with three concentration levels, and each level was set with three parallel levels, three repetitions for each parallel. For the determination of total phenol content and antioxidant capacity, there were three parallels for each level of the experimental sample and three repetitions for each parallel.

According to previous studies [35,36], the existence of epidermis can effectively prevent PAHs from penetrating into biological tissues from flue gas during the roasting of meat products. In particular, high molecular PAHs can be effectively intercepted by epidermis without being transferred into physical tissues. Only small amounts of naphthalene and acenaphthene with low molecular weight and low toxicity have been detected in roast duck meat, and no carcinogenic substances have been detected, significantly different from duck skin samples. Therefore, in this study, duck meat will not be investigated.

### 2.3. Roast Duck Processing and Curing Process

Raw duck was processed according to traditional techniques. Rinsed duck was submerged in the prepared pickling solution, with the duck breast facing upward, ensuring that the floating duck was fully immersed in the feed water for about 12 h. Air was pumped into the duck under the skin for inflation, making the duck look appealing. Then, boiling water (100 °C) was poured onto the duck surface three times. After blanching, the duck was dried at 2–4 °C for 2–4 h, and then caramel and maltose were applied to the duck to induce acceptable color change before freezing. Subsequently, the duck was frozen at −5 °C for 2–3 days to make the duck skin thicker and more delicious. Before roasting, the duck was thawed at 2–4 °C and heated to room temperature. Finally, the duck was hung in the traditional oven (Gua-lu) for roasting by an experienced roast duck chef. The top and bottom temperatures of the oven were set at 202–203 °C, and the center temperature was 224 °C during roasting. The total duck roasting time was 45–60 min.

After roasting, the duck skin was separated from the duck breast and duck leg, and the skin was ground and homogenized for PHA detection.

### 2.4. Determination of Total Phenolic Content and Antioxidant Capacity of Natural Extracts

#### 2.4.1. Total Phenol Content in Natural Extracts

Total phenolic content was determined using Folin–Ciocalteu agent according to Wang et al. [27], and 0.01 ± 0.001 g gallic acid reference substance was accurately weighed with an analytical balance and placed in a 10 mL volumetric flask. An appropriate amount of methanol was added for ultrasonic dissolution, and constant-volume pure water was added to the scale. The flask was shaken to obtain 1 mg/mL standard gallic acid reserve solution. Standard curve was prepared using gallic acid solution at 10, 20, 30, 40 and 50 μg/mL.

Volumes of 1 mL gallic acid working solution, water (blank) and sample solution were tested respectively. A total of 0.5 mL Folin reagent, 2 mL 7.5% sodium carbonate solution and 6.5 mL water were added, and the mixture was oscillated for 1 min and reacted in 70 °C water for 30 min. The absorbance was measured at 750 nm (A). While 1 mL distilled water was used as the blank, the absorbance of the gallic acid working solution (A) was used as the ordinate, and the gallic acid concentration was used as the abscissa to draw a standard curve. The absorbance value (A) of the sample was located within the appropriate range of the standard curve, and the total phenolic content of the liquid to be tested was calculated according to the standard curve. The result was expressed as mg gallic acid equivalent (GAE) in 1 g of dry matter (mg GAE/g).

#### 2.4.2. Trolox Equivalent Antioxidant Capacity (TEAC)

In the TEAC method, Trolox is used as the standard antioxidant and serves as a reference for the total antioxidant capacity of other antioxidants. Trolox is an analog of vitamin E and has antioxidant capacity similar to vitamin E. Data obtained by this method are called the TEAC. Following the method of Ozgen et al. [37], ABTS radical cation (ABTS) was produced by reacting 5 mL of 7 mmol/L ABTS aqueous solution with 88 μL of 140 mmol/L potassium persulfate (K_2_S_2_O_8_) aqueous solution, covering the mixture with foil and leaving it for 24 h at room temperature. Trolox aqueous solution in the linear range of 50–1000 μmol/L was used to plot a calibration curve, and ultra-pure water was used as the blank control. Following the method of Re et al. [38], 1 mL of ABTS solution was placed in a 15 mL centrifuge tube, and 20 μL of Trolox or sample extract solution was added and oscillated evenly. Then, the reaction was carried out in 30 °C water for 6 min. The absorbance of the solution was measured at 734 nm with a UV spectrophotometer. The results were expressed as the ratio of ABTS radical scavenging ability (change in absorbance) of a plant extract to the scavenging ability of standard antioxidant Trolox, in units of TEAC. That is, the scavenging capacity of ABTS per gram of Trolox was equivalent to the number of mg (mmol/g of Trolox).

### 2.5. The Determination of PAHs

Referring to the “National food safety standard for the use of food additives” and the ingredient ratio of the “vegetable crispy” roast duck curing process in the cheap square(Beijing), and also considering that the extracts may not have been fully dispersed throughout the sample, it was determined that a duck needs about 4 kg of pickling liquid for curing. Thus, the plant extracts were added to the curing solution according to the proportion of 10 g, 30 g and 50 g per 4 kg of pickling liquid, which has the antioxidant effect by penetrating into the raw duck. In this experiment, four natural plant extracts with antioxidant activity (GSE, RE, EBL and GTE) were added to the pickling liquid at three levels (5, 15 and 25 g/kg) to assess their effects on the formation of 16 PAHs in duck skin. Each level was set in three parallels and three repeats were conducted for each parallel. 

#### 2.5.1. Determination of the PAH Content in Roast Duck

Following Surma [39], a sample of 4 g (accurate to 0.01 g) was homogenised with 10 mL ethyl acetate extract for 1 min, vortexed, treated with ultrasonic extraction for 20 min with 1–2 g anhydrous magnesium sulfate added and then centrifuged at 5000 r/min for 5 min. The supernatant was transferred to a 100 mL round-bottom flask. Then, 10 mL ethyl acetate was added to the original glass centrifuge tube, and centrifugation was repeated. After centrifugation, the supernatant was merged into the above round-bottom flask. The supernatant was concentrated in a 35 °C water bath at 110 r/min by rotary evaporation until it was nearly dry. Gel permeation chromatography (GPC) purification was performed after it passed a 0.45 µm organic needle filter. During the experiment, a blank sample was added for each batch of samples, 10 mL ethyl acetate replaced the duck skin sample for purification and GPC used Bio Beads SX-3 gel column of styrene resin. The absorbance of the solution was measured at 254 nm. The mobile phase consisted of ethyl acetate/cyclohexane (1:1, *v/v*). Components were collected in 22–45 min in a 250 mL round-bottom flask, and the solution was concentrated to about 2 mL in a water bath at 35 °C by rotary evaporation at 100 r/min. Nitrogen was blown until nearly dry, and the mixture was redissolved in 1 mL ethyl acetate cyclohexane (1:1, *v/v*) through 0.22 µm organic filter membrane.

Chromatography was performed using a DB-5 capillary column (30 m × 0.25 mm × 0.25 μm). The column temperature procedure was as follows: initial temperature at 70 °C for 2 min, increased to 150 °C at 25 °C/min, 200 °C at 3 °C/min and 280 °C at 8 °C/min and then kept for 15 min. The procedure lasted 46.87 min in total. Injection was performed under the following conditions: injector temperature of 290 °C; no shunt injection (1 min); injection volume of 1 μL; flow control mode of constant pressure (97.9 kPa); helium carrier gas purity ≥99.999%. The mass spectrometry (MS) reference conditions were as follows: electron bombardment ion source (EI), EI ionization energy of 70 eV, ion source temperature of 230 °C; interface temperature of 280 °C; solvent cutting time of 3 min; data acquisition mode of selected ion monitoring mode (SIM).

#### 2.5.2. The Validation of the PAH Content Determination

In this part, the contents of PAHs in roast duck skin were verified. The method was validated by the parameters of the limit of detection (LOD), limit of quantification (LOQ), linearity, recovery and precision. Under the above experimental conditions, the LOD and LOQ of this method were determined by signal-to-noise (S/N) ratios of no less than 3 and 10, respectively. By plotting the relationship between the peak area and the mass concentration, quantitative curves were constructed for the PAHs in the concentration range of 1, 5, 10, 20, 50, 100 and 200 μg/kg, and the linear equation, linear range and correlation coefficient were obtained. In order to determine the accuracy of this method, recovery tests were carried out. Mixed standard solutions were added to blank samples at 1, 5 and 10 μg/kg for the recovery experiments. The average recovery rate and relative standard deviation (RSD) were determined.

### 2.6. Data Processing

All experimental samples were set in 3 parallels, and each parallel was set in triplicate. The results were statistically analyzed by ANOVA (*p* < 0.05). The mean values were compared using Duncan’s test. Statistical analyses were all performed with SPSS 17.0. The data were expressed as means ± SD (*n* = 3). Origin 8.5 was employed to draw the diagrams.

## 3. Results and Discussion

### 3.1. The Total Phenolic Contents and Antioxidant Capacities of the Four Natural Extracts

#### 3.1.1. The Total Phenolic Contents of the Four Natural Extracts

Before pickling with the four extracts at a concentration of 15 g/kg, the polyphenol content of the pickling solution was determined three times in parallel. The Folin–Ciocalteu colorimetry results are shown in Figure 1. The highest total phenol content was in GTE at 277 mg GAE/g, which was significantly higher than those in the other extracts. The main effective component of GTE is tea polyphenols (TP), accounting for 20–30% of the dry weight of tea [20,26,27]. The total phenol contents of GSE and EBL were 205 and 195 mg GAE/g, respectively, and there was no significant difference between them. RE had the lowest total phenol content among the four extracts, which was 85 mg GAE/g, only about 1/3 of that of GTE.

#### 3.1.2. The Antioxidant Capacities of Four Natural Extracts

Single electron transfer (SET) and hydrogen atom transfer (HAT) are two key mechanisms for understanding the antioxidant activity or free radical scavenging characteristics of these natural extracts [40]. TEAC assay aims to measure SET in the antioxidative process, and the results are expressed as the TEAC [41]. When antioxidants were added, the stock solution color lightened. In this experiment, the reduction in the light absorption value of the pickling solution after the extract addition was measured to obtain the free radical scavenging abilities of the four antioxidants. Figure 2 shows that under the same addition concentration, the TEACs of GTE, GSE and EBL were not significantly different and were significantly higher than that of RE, which was basically consistent with the polyphenol contents of the four extracts. The comprehensive evaluation results showed that the polyphenol content and total antioxidant capacity of GTE were the highest, followed by EBL, GSE and RE. The TP in GTE has the function of scavenging reactive oxygen free radicals and is an excellent natural antioxidant [20,26,27]. Bamboo leaf flavone is the main active component in EBL, which has a function similar to superoxide dismutase (SOD) and glutathione peroxidase (GSH Px) [28,29,30]. GSE has significant antioxidant activity and free radical scavenging ability. The antioxidant capacity of proanthocyanidins in the extract is 20 times that of vitamin E and 50 times that of vitamin C [18,19,20]. As an antioxidant, RE mainly inhibits oil oxidation in food by terminating free radicals and inhibiting singlet oxygen generation [33].

### 3.2. The Effects of Natural Extracts on the Formation of PAHs in Roast Duck

#### 3.2.1. Validating the Method of PAH Determination

The accuracy of the detection method was verified by LOD, LOQ, linearity, recovery and precision. Under the above experimental conditions, the LOD and LOQ of the method were determined by S/N ratios of more than 3 and 10, respectively. LOD ranged from 0.08 to 0.39 μg/L, and LOQ ranged from 0.25 to 1.29μg/L. In this experiment, the correlation coefficient R^2^ was between 0.996 and 0.999. To verify the accuracy of the method, a total of six experiments were carried out to determine the average recovery and RSD. The average recoveries at the concentrations of 1, 5 and 10 μg/kg were 66.2–108.1% with RSDs of less than 15%. The results met the requirements for the detection and recovery of trace substances stipulated in the standards “Criterion on quality control of laboratories—Chemical testing of food”.

#### 3.2.2. The Effects of Natural Extracts on ∑16 PAH Levels

More than 100 kinds of PAHs have currently been identified. The 16 most well-known PAHs proposed by the U.S. Environmental Protection Agency (USEPA) are naphthalene (Nap), acenaphthene (Anl), acenaphthene (Ane), fluorene (Flu), phenanthrene (Phen), anthracene (Ant), fluoranthene (Flt), pyrene (Pyr), benz [a] anthracene (BaA), chrysene (Chr), benzo [b] fluoranthene (BbF), benzo [k] fluoranthene (BkF), benzo [a] pyrene (BaP), indene [1,2,3-cd] pyrene (InP), dibenzo [a,h] anthracene (DahA) and benzo [g,h,i] perylene (BghiP). The EU and China have no regulations regarding the total acceptable amounts of these 16 PAHs (∑16 PAHs) in smoked and roasted meat products.

The PAH content in duck skin is significantly higher than that in duck meat. The concentration of ∑16 PAHs in roast duck skin without any extracts (control group) was 288.43 μg/kg, which was significantly higher than that in roast duck meat (241.78 μg/kg). This was consistent with the discovery by Lin et al. [42]. The content of ∑16 PAHs in duck skin in the pickled group was compared with that in the control group, and the results are shown in Table 1. Overall, the average content of the 16 PAHs in the duck skin in the control group was 288.4 ng/g, and that of ∑16 PAHs in the duck skin in all the pickled groups was significantly different from that in the control group. The results indicated that adding a certain amount of GTE, EBL, RE or GSE to the pickle solution can effectively reduce the content of ∑16 PAHs, and the content showed a gradually decreasing trend with increasing concentrations of the extracts. When the addition concentration of the extracts was 5 g/kg, the order of the content of ∑16 PAHs in the four pickling groups with extracts was GSE < GTE < RE < EBL (low to high). There was no significant difference between EBL and RE, and there was no significant difference between GTE and GSE. When the concentration of the extracts was 15 g/kg, the content of ∑16 PAHs in roast duck skin in the four groups added with extracts exhibited significant differences. The order of the total content of ∑16 PAHs in the four experimental groups was EBL < GSE < GTE < RE. When the addition concentration of the extracts was 25 g/kg, compared with the control group, the content of ∑16 PAHs in duck skin cured with the four extracts decreased most significantly. The higher the addition concentration of the extract, the more obvious the reduction in the content of ∑16 PAHs.

#### 3.2.3. The Effects of Natural Extracts on the PAH4 Level

PAH4, first proposed by the European Food Safety Authority (EFSA) in 2008, refers to the sum of the contents of four PAHs including BaP, BaA, BbF and CHR. EFSA claimed that a single class of PAHs was not suitable enough to be used as a marker; instead, PAH4 could suffice as a marker based on data related to the frequency of formation and toxicity [43]. Subsequently, the European Union (EU) regulated the content of PAHs in food through Regulation 835/2011 [44,45]. The maximum residues of BaP and PAH4 in smoked meat and bacon products were specified as 2.0 μg/kg and 12 μg/kg, respectively [46].

The contents of PAH4 in the marinated groups were compared with that in the control group, and the results are shown in Table 1. Overall, the contents of PAH4 in duck skin and duck meat far exceeded 12 μg/kg, with averages of 44.61 μg/kg and 28.71 μg/kg, respectively, exceeding the EU’s maximum residue limit for smoked meat and bacon products. Compared with the control group, the addition of four natural plant extracts to the marinade reduced the total content of PAH4 in roast duck. With the increase in the extract concentration, the total content of PAH4 in roast duck showed a gradual decrease.

When the concentration was 5 g/kg, the total contents of PAH4 in duck skin in the GTE, EBL and GSE groups were significantly different from that in the control group (*p* < 0.05), while there was no significant difference between RE and the control group (*p* > 0.05). When the concentration was 15 g/kg, the total contents of PAH4 in the duck skin in the four extract groups were significantly different from that in the control group. Among them, the total contents of PAH4 in duck skin in the GTE and EBL groups were lower than the EU limit of 12 μg/kg. When the extract concentration was 25 g/kg, the four extracts could effectively reduce the total content of PAH4. The EBL group had better performance than the other three extracts in the duck meat samples. We detected only small amounts (1.93–2.15 μg/kg) of PAH4, and there was no significant difference between the concentrations (*p* > 0.05). 

#### 3.2.4. Effects of Natural Extracts on BaP

Among many PAHs, BaP was the first environmental chemical carcinogen found. It has strong carcinogenicity, a wide distribution range, stable properties and a certain correlation with other PAHs that can be detected using sensitive detection methods. Therefore, it is often used as an indicator compound. China has formulated mandatory food safety standards regarding the limits of PAHs, taking BaP as the indicator to evaluate the pollution of food by PAHs, and the limit of BaP in meat and meat products (smoked and roast meat) is 5.0 μg/kg.

The BaP content in marinated duck skin was compared with a control group, and the results are shown in Table 1. Overall, the content of BaP in duck skin in the control group was significantly higher than 5.0 μg/kg, reaching 7.4 μg/kg, which exceeded the Chinese limit of BaP in food (*p* < 0.05). The addition of the four plant extracts to the pickling solution inhibited the formation of BaP in duck skin. When the addition concentration was 5 g/kg, the content of BaP in duck skin in GTE and EBL groups decreased significantly, and the content of BaP in samples pickled with GTE was lower than 2.0 μg/kg. There was no significant difference in the content of BaP in duck skin in the GSE and RE groups compared with the control group. When the addition concentration of the experimental groups was 15 g/kg, the content of BaP in the duck skin in all the test groups was significantly lower than that in the control group. Except for the RE group, the BaP content in the other three experimental groups was below 5.0 μg/kg. When the concentration of BaP was 25 g/kg, the content of BaP in all treated duck skin samples could meet the limit standard of BaP in meat and meat products (smoked and roasted meat).

### 3.3. The Analysis of the PAH Inhibition Rate by Four Natural Extracts

#### 3.3.1. The Analysis of the Inhibition Rate of the 16 PAHs

To compare the inhibition effects of four natural extracts on PAHs, the inhibition rates of the four natural extracts on PAHs were calculated respectively. The inhibitory rate was determined according to the equation:(1)Inhibitory rate (%) = (Ac − At)/Ac × 100

Ac was the total amount of PAHs in control samples (g/kg), and At was the total amount of PAHs in natural extracts added to roast duck (g/kg).

The total content of 16 PAHs in roast duck skin without extracts was used as the control, and the samples with added extracts were used as the experimental groups to calculate the percentage decreases in the total content of 16 PAHs. Then, we could obtain the inhibition rate of 16 PAHs at the concentration levels of 5, 15 and 25 g/kg for the four extracts. The results in Figure 3 show that all of the four natural extracts have a good inhibitory effect on the 16 PAHs. As the concentration of the extracts increases, the inhibition rate increases as well. Each extract has different performance on the inhibition rate for the 16 PAHs. In the duck skin samples, when the addition level was 5–25 g/kg, the ranges of the inhibition rates of GTE, EBL, RE and GSE were 44.18–61.45%, 28.72–66.60%, 34.42–67.12% and 46.60–78.74%, respectively. According to the results, the antiradical activity was not directly correlated with PAH formation inhibition. There were similar findings in other studies [27]. This may be attributed to the free radical scavenging abilities of different antioxidants, which may be dependent upon their molecular structures, especially the effective reactivity of phenols and polyphenols with free radicals [21,47]. As a result, the effective reactivity of polyphenols in GTE with free radicals was lower than that in GSE despite the highest antioxidant capacity and polyphenol content in GTE.

#### 3.3.2. The Analysis of the Inhibition Rate of PAH4

Taking the total content of PAH4 in roast duck skin without extract as the control and the samples with extract as the experimental groups, the decrease percentage of the PAH4 content after extract addition was calculated. Figure 4 summarizes the inhibition rates of the four natural extracts on PAH4. It can be seen from the figure that the inhibitory effect on PAH4 in duck skin gradually improved with the increase in the concentration of the four natural extracts. When the addition level was 5–25 g/kg, the rates of GTE, EBL, RE and GSE inhibition of PAH4 were 58.1–79.7%, 40.5–78.0%, 0.9–43.6% and 5.4–72.7%, respectively. The inhibitory effects of GTE, EBL and GSE on PAH4 were significantly higher than that of RE, which may be because the polyphenol content and antioxidant capacity of RE were significantly lower than those of the other three.

#### 3.3.3. The Analysis of the Inhibition Rate of BaP

The addition of four plant extracts to the pickling solution has certain inhibitory effect on the formation of BaP in duck skin, which can be seen in Figure 5. When the concentration was 5 g/kg, the content of BaP in duck skin in the GTE and EBL groups was significantly reduced, and the content of BaP in the GTE group was less than 2.0 μg/kg; the contents of BaP in the duck skin in the GSE and RE groups were not significantly different from that in the control group. When the concentration of the pickled groups was 15 g/kg, the contents of BaP in the duck skin in all the pickled groups were significantly lower than that in the control group. Except for the RE group, the BaP contents in the pickled samples were below 5.0 μg/kg. When the concentration was 25 g/kg, the BaP contents in all marinated duck skin samples met the limit for BaP in meat and meat products (smoked, roasted and barbecued) in China. When the effects of the four extracts on BaP were compared, the inhibitory effect of GTE on BaP was the best at all three concentrations, which may be attributable to the higher polyphenol content and antioxidant activity of GTE than the other three. Viegas et al. [21] found that GTE rich in TP could effectively reduce or inhibit the production of some HCAs and PHAs in roast meat products [20,26,27]. Relevant studies have shown that RE can inhibit the formation of heterocyclic amine PhIP in roast beef patties, but few studies have demonstrated the effects of RE on the formation of PAHs [32,33,34]. Studies on the effect of beer pickling on PAHs in charcoal-roasted pork concluded that the inhibitory effects of beer marinades on PAHs increased as their radical-scavenging activities increased.

## 4. Conclusions

Among the four plant extracts selected in the experiment, GTE had the highest total phenol content and antioxidant capacity, while RE had the lowest. In addition, the inhibition rates of the PAHs showed that the four plant extracts could reduce BaP in roast duck skin to a certain extent, It is worth noting that the inhibition rates of GTE on BaP and PAH4 in roast duck skin were the highest, while GSE had the best inhibition effect on 16 PAHs. The cause may be that GSE contains special substances with better inhibitory effects on 16 PAHs than the other three extracts. In conclusion, adding natural antioxidants during the curing of roast duck can reduce the contents of PAHs in roast duck. The antioxidant capacity and polyphenol content were key factors affecting the formation of PAHs in roast duck, which provided a theoretical basis for follow-up studies on the control of PAHs in roast duck. Moreover, compared with the traditional pickling solution, the addition of four natural extracts may bring some special flavor to the roast duck, which can be used as a research direction in the future.

## Figures and Tables

**Figure 1 foods-11-02104-f001:**
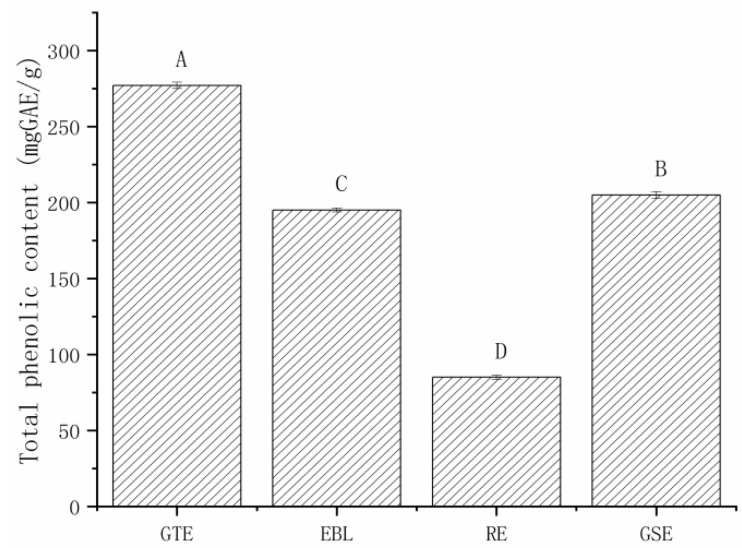
The total phenolic contents of four extracts. Note: GTE: pickling solution with green tea extract; EBL: pickling solution with extract of bamboo leaves; GSE: pickling solution with grape seed extract; RE: pickling solution with rosemary extract. Different letters (A, B, C and D) indicate significant differences (*p* < 0.05).

**Figure 2 foods-11-02104-f002:**
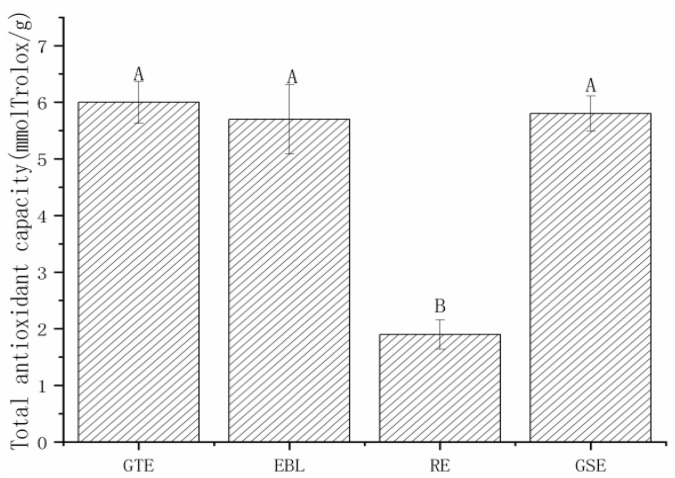
The total antioxidant capacities of four extracts (by TEAC assay). Note: GTE: pickling solution with green tea extract; EBL: pickling solution with extract of bamboo leaves; GSE: pickling solution with grape seed extract; RE: pickling solution with rosemary extract. The same letters (A and B) indicate that there is no significant difference (*p* < 0.05).

**Figure 3 foods-11-02104-f003:**
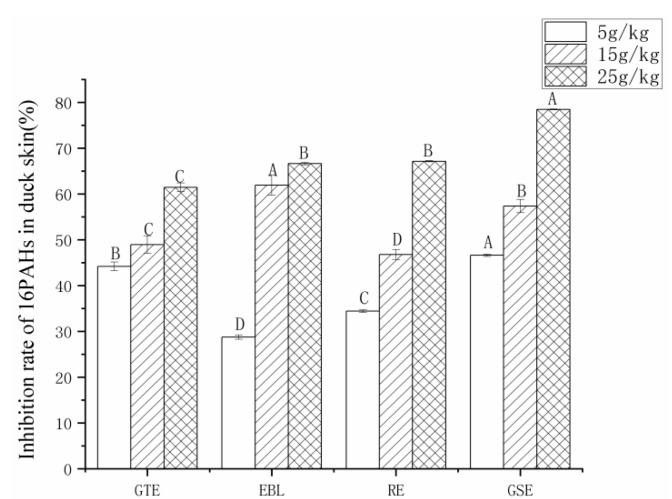
The inhibition rates of 16 PAHs in roast duck skin treated with four different extracts and three levels of addition. Note: GTE: pickling solution with green tea extract; EBL: pickling solution with extract of bamboo leaves; GSE: pickling solution with grape seed extract; RE: pickling solution with rosemary extract. Different letters (A, B, C and D) indicate significant differences comparison among extracts (*p* < 0.05).

**Figure 4 foods-11-02104-f004:**
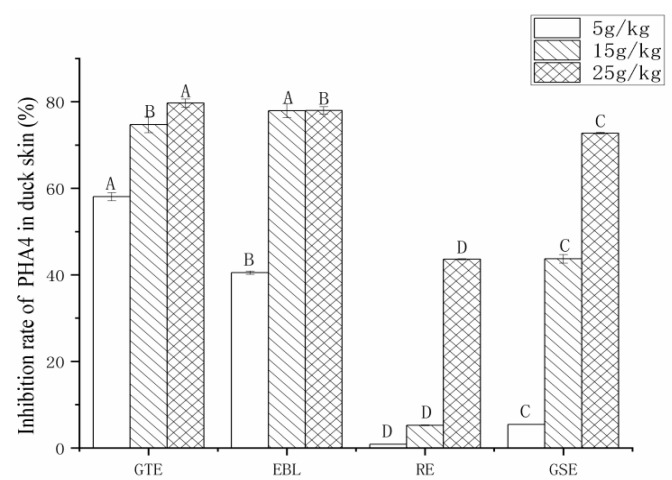
The total content of PAH4 in roast duck skin treated with four different extracts and three levels of addition. Note: GTE: pickling solution with green tea extract; EBL: pickling solution with extract of bamboo leaves; GSE: pickling solution with grape seed extract; RE: pickling solution with rosemary extract. Different letters (A, B, C and D) indicate significant differences comparison among extracts (*p* < 0.05).

**Figure 5 foods-11-02104-f005:**
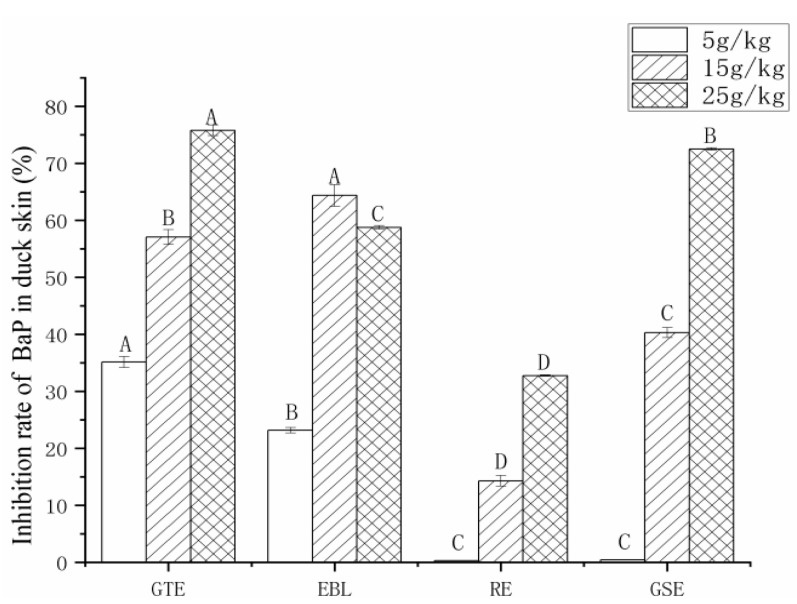
Contents of BaP in roast duck skin treated with four different extracts and three levels of addition. Note: GTE: pickling solution with green tea extract; EBL: pickling solution with extract of bamboo leaves; GSE: pickling solution with grape seed extract; RE: pickling solution with rosemary extract. Different letters (A, B, C and D) indicate significant differences comparison among extracts (*p* < 0.05).

**Table 1 foods-11-02104-t001:** PAHs contents of in roast duck skin prepared with four natural extracts.

Compounds	Control Group	PHAs (μg/kg)											
		GTE 5 g/kg	GTE 15 g/kg	GTE 25 g/kg	EBL 5 g/kg	EBL 15 g/kg	EBL 25 g/kg	RE 5 g/kg	RE 15 g/kg	RE 25 g/kg	GSE 5 g/kg	GSE 15 g/kg	GSE 25 g/kg
Nap	24.78 ± 0.86 ^cd^	25.34 ± 2.61 ^c^	25.79 ± 2.11 ^bc^	21.84 ± 2.52 ^d^	24.69 ± 0.37 ^cd^	22.23 ± 0.90 ^d^	16.10 ± 1.63 ^e^	31.00 ± 1.08 ^a^	13.53 ± 2.61 ^e^	9.54 ± 0.70 ^f^	14.59 ± 0.54 ^e^	3.33 ± 1.06 ^g^	1.26 ± 0.47 ^g^
Anl	14.93 ± 0.17 ^b^	nd	nd	nd	17.83 ± 0.93 ^a^	8.82 ± 1.87 ^c^	5.20 ± 0.11 ^de^	5.74 ± 0.66 ^d^	5.70 ± 0.81 ^d^	1.95 ± 0.58 ^f^	4.91 ± 0.12 ^de^	4.31 ± 0.35 ^e^	nd
Ane	3.84 ± 0.43 ^a^	0.75 ± 0.06 ^ef^	0.60 ± 0.09 ^fg^	0.36 ± 0.07 ^gh^	2.38 ± 0.14 ^b^	1.69 ± 0.17 ^c^	1.62 ± 0.56 ^c^	1.08 ± 0.15 ^de^	1.45 ± 0.20 ^cd^	nd	1.41 ± 0.12 ^cd^	1.62 ± 0.05 ^c^	nd
Flu	22.64 ± 1.23 ^a^	10.37 ± 0.67 ^c^	5.87 ± 0.73 ^e^	10.09 ± 0.38 ^c^	11.91 ± 0.66 ^b^	8.19 ± 0.75 ^d^	8.84 ± 0.65 ^d^	4.02 ± 0.06 ^f^	3.84 ± 0.34 ^fg^	1.10 ± 0.36 ^i^	5.44 ± 0.54 ^e^	5.28 ± 0.72 ^e^	2.81 ± 0.24 ^gh^
Phen	91.92 ± 2.43 ^a^	58.01 ± 1.72 ^b^	57.06 ± 5.22 ^bc^	45.05 ± 3.61 ^d^	53.16 ± 1.39 ^c^	33.05 ± 3.45 ^e^	33.47 ± 3.50 ^e^	29.37 ± 0.64 ^ef^	26.38 ± 1.90 ^f^	14.80 ± 0.81 ^g^	31.75 ± 1.10 ^e^	29.57 ± 1.35 ^ef^	15.67 ± 0.96 ^g^
Ant	16.27 ± 0.72 ^a^	9.75 ± 0.57 ^c^	9.71 ± 0.58 ^c^	7.13 ± 0.32 ^de^	10.84 ± 0.52 ^b^	6.31 ± 0.63 ^ef^	6.23 ± 1.01 ^ef^	8.05 ± 0.07 ^d^	6.03 ± 0.57 ^fg^	3.05 ± 0.45 ^h^	6.81 ± 0.87 ^ef^	6.72 ± 0.29 ^ef^	5.20 ± 0.36 ^g^
Flt	31.10 ± 2.41 ^a^	20.73 ± 0.94 ^c^	20.69 ± 0.86 ^c^	11.62 ± 2.28 ^e^	27.88 ± 2.02 ^b^	10.62 ± 2.71 ^e^	6.71 ± 1.35 ^f^	25.54 ± 0.60 ^b^	19.54 ± 2.70 ^c^	15.18 ± 0.47 ^d^	18.64 ± 0.49 ^c^	18.67 ± 0.80 ^c^	11.79 ± 0.30 ^e^
Pyr	28.52 ± 1.78 ^a^	13.99 ± 1.80 ^e^	13.40 ± 1.59 ^e^	3.29 ± 0.65 ^g^	25.71 ± 0.78 ^b^	7.17 ± 1.02 ^f^	7.24 ± 0.15 ^f^	25.15 ± 0.43 ^bc^	23.29 ± 2.40 ^c^	17.76 ± 0.49 ^d^	16.56 ± 0.56 ^d^	16.34 ± 1.28 ^d^	8.74 ± 0.44 ^f^
B[a]A	9.45 ± 1.24 ^a^	6.13 ± 0.21 ^bc^	4.06 ± 0.17 ^de^	2.29 ± 0.50 ^f^	7.26 ± 0.76 ^b^	3.37 ± 0.93 ^ef^	6.77 ± 1.88 ^b^	9.41 ± 0.45 ^a^	9.39 ± 1.31 ^a^	5.24 ± 0.46 ^cd^	9.19 ± 0.28 ^a^	6.32 ± 0.65 ^bc^	3.02 ± 0.35 ^ef^
Chry	18.77 ± 0.98 ^a^	9.16 ± 0.67 ^e^	3.81 ± 0.30 ^g^	4.12 ± 0.42 ^fg^	10.56 ± 0.43 ^d^	2.81 ± 0.26 ^h^	nd	18.51 ± 0.59 ^a^	17.86 ± 0.32 ^b^	9.73 ± 0.37 ^e^	16.73 ± 0.25 ^c^	9.15 ± 0.30 ^e^	4.75 ± 0.37 ^f^
B[b]F	8.97 ± 0.23 ^a^	2.17 ± 0.24 ^d^	2.04 ± 0.36 ^d^	2.24 ± 0.43 ^d^	4.17 ± 0.35 ^c^	0.52 ± 0.34 ^e^	nd	8.90 ± 0.13 ^a^	8.67 ± 0.55 ^a^	5.19 ± 0.23 ^b^	8.88 ± 0.14 ^a^	5.23 ± 0.30 ^b^	2.34 ± 0.42 ^d^
B[k]F	9.82 ± 0.96 ^a^	3.35 ± 0.47 ^d^	2.84 ± 0.62 ^d^	2.76 ± 0.41 ^d^	4.64 ± 0.41 ^c^	1.86 ± 0.36 ^e^	1.10 ± 0.10 ^e^	9.98 ± 0.03 ^a^	7.29 ± 0.61 ^b^	4.82 ± 0.30 ^c^	7.01 ± 0.19 ^b^	7.34 ± 0.80 ^b^	4.57 ± 0.48 ^c^
B[a]P	7.42 ± 0.67 ^a^	1.25 ± 0.45 ^de^	1.37 ± 0.45 ^de^	0.42 ± 0.27 ^ef^	4.57 ± 0.47 ^b^	3.15 ± 0.85 ^c^	3.06 ± 0.25 ^c^	7.40 ± 0.93 ^a^	6.36 ± 0.71 ^a^	4.99 ± 0.67 ^b^	7.39 ± 0.52 ^a^	4.43 ± 0.74 ^b^	2.04 ± 0.42 ^d^
InP	nd	nd	nd	nd	nd	nd	nd	5.00 ± 0.02 ^a^	4.20 ± 0.15 ^c^	1.47 ± 0.28 ^d^	4.70 ± 0.17 ^b^	4.72 ± 0.34 ^b^	nd
B[g,h,i]P	nd	nd	nd	nd	nd	nd	nd	nd	nd	nd	nd	nd	nd
D[ah]A	nd	nd	nd	nd	nd	nd	nd	nd	nd	nd	nd	nd	nd
16 PAHs	288.43 ± 1.82 ^a^	160.99 ± 3.34 ^c^	147.24 ± 6.70 ^d^	111.20 ± 3.42 ^f^	205.59 ± 0.98 ^b^	109.79 ± 5.64 ^f^	96.33 ± 1.70 ^g^	189.15 ± 0.38 ^b^	153.53 ± 3.87 ^c^	94.84 ± 1.01 ^g^	154.01 ± 3.73 ^c^	123.03 ± 5.09 ^e^	62.18 ± 0.95 ^h^
PAH4	44.61 ± 1.94 ^a^	18.71 ± 1.29 ^d^	11.28 ± 0.95 ^ef^	9.06 ± 1.01 ^g^	26.55 ± 1.45 ^c^	9.85 ± 1.01 ^fg^	9.83 ± 1.69 ^fg^	44.22 ± 0.75 ^a^	42.28 ± 1.37 ^b^	25.16 ± 0.36 ^c^	42.19 ± 0.74 ^b^	25.12 ± 0.86 ^c^	12.16 ± 0.58 ^e^

Note: GTE: pickling solution with green tea extract; EBL: pickling solution with extract of bamboo leaves; GSE: pickling solution with grape seed extract; RE: pickling solution with rosemary extract. Results are presented as the mean ± standard deviation, *n* = 9. Means with different letters in the same row show significantly different (*p* < 0.05). nd: not detected or below the limit of quantification (LOQ).

## Data Availability

The data presented in this study are available on request from the corresponding author.

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
