# Peer review of "The Effects of Different Natural Plant Extracts on the Formation of Polycyclic Aromatic Hydrocarbons (PAHs) in Roast Duck"

_foods, 2022, doi:10.3390/foods11142104_

Round 1

Reviewer 1 Report

The manuscript reports research into effects of different natural plant extracts on the formation of PAHs in roast duck. The subject falls into the scope of the journal. However, several issues should be addressed. Therefore, the manuscript can not be accepted in the present form. The acceptance of manuscript could be considered if the manuscript is revised and corrected according to the following comments, and after that, the content of the corrected version is interesting and valuable.

 TITLE:

You must indicate the meaning of the acronym “PAHs” in title.

 ABSTRACT:

-          According FOODS’ instructions, the abstract should be a total of about 200 words maximum. Therefore, you should reduce the abstract.

-          Write a clear objective in the abstract.

-          Line 21: you must indicate the meaning of the acronym PAH4 the first time it appears in the text.

-          Line 22: space between number and unit, that means, “75.8 %” instead of “75.8%”. This rule is applicable across manuscript.

 INTRODUCTION:

Line 32: you have used superscript for references when you don’t have to. Therefore, correct the format of all references across the text.

Lines 69-70: “et al” should be in italic.

Line 70: space between number and unit (180 ºC). This rule is applicable across manuscript.

Lines 75-78:

-          Objective should be rewrite, it is not clear.

-          Why have you used references in the objective?

-          You do not have to use references in the objective.

 MATERIAL AND METHODS:

You should write a section with the design of the experiment at the beginning of “Material and methods”.

Line 86: water is liquid. Why do you use “g” for water instead of “L”?

Lines 82 and 85: you have to write the city and country of the companies.

Section “2.3.1. Total phenol content in natural extracts”: you should write the reference of the method.

Line 128: “et al” should be in italic.

Lines 142-146:

-          Why have used these doses?

-          The doses are g/kg, kg of what? Because you have added the plant extract to the pickling liquid.

-          Why did you assess the effects on the formation of PAHs in duck skin? Do people eat the skin? In addition to this, aren’t there PAHs in duck?

-          You have to answer my questions and introduce the responses in that paragraph.

-          You have to rewrite that paragraph that is not clear.

Section “2.4.1. Determination of the PAH content in roast duck”

-          You should write the reference of the method used.

-          Line 148: sample of what? Roast duck or duck skin?

-          You must indicate the meaning of the acronym MS the first time it appears in the text.

 Section “2.5. Data processing”:  you should explain the ANOVA and test to detect differences among samples.

 RESULTS AND DISCUSSION:

Figures 1, 2, 3, 4, 5:

-          You should define the acronyms of the samples again, because the figures should be self-explanatory.

-          You should explain the meaning of different letters in the figure (significant differences?)

Table 1:

-          You should define the acronyms of the samples again, because the tables should be self-explanatory.

-          You should explain the meaning of different letters in the table, significant differences among extracts or doses?

-          Indicate the units used.

Figures 1, 2, 3, 4, 5:

CONCLUSION:

Lines 384-395: you should remove these lines because they are a summary of results and they are not conclusions.

Lines 401-402: the dose added is not small and could affect the flavor. In addition to this, you write “the flavor changes … can be investigated in future research”. Thus, you are incoherent and you are confirming that the flavor will change. You should rewrite these lines.

 REFERENCES:

This section should be corrected according to the FOODS journal’s instructions. There are too many mistakes in the references.

Author Response

Dear Reviewer:

Thank you for your comments on our manuscript, which are very helpful to improve our manuscript’s quality. We have made modification according to your suggestions. Please see the revised version uploaded for details. Below are the responses to comments.

(1)You must indicate the meaning of the acronym “PAHs” in title.

 Response: Thank you for your comment. We have added the meaning of the acronym “PAHs” in title. Please check.

(2) According FOODS’ instructions, the abstract should be a total of about 200 words maximum. Therefore, you should reduce the abstract.

Response: Thank you for pointing out this issue. We have reduced the number of abstract to less than 200 words. Please check.

(3)    Write a clear objective in the abstract.

Response: Thank you for your comment. We have modified the details and rewritten the objective in the abstract.

(4) Line 21: you must indicate the meaning of the acronym PAH4 the first time it appears in the text.

 Response: Thank you for your comment. We have added the meaning of the acronym “PAH4” in abstract. Please check.

(5)  Line 22: space between number and unit, that means, “75.8 %” instead of “75.8 %”. This rule is applicable across manuscript.

Response: Thank you for your comment. We have modified the details. Please check.

(6)Line 32: you have used superscript for references when you don’t have to. Therefore, correct the format of all references across the text.

Response: Thank you for your comment. We have modified the details.

(7)Lines 69-70: “et al” should be in italic.

Response: Thank you for your comment. We have modified the details. Please check.

(8)Line 70: space between number and unit (180 ºC). This rule is applicable across manuscript. Please check.

Response: Thank you for your comment. We have modified the details.

(9)Lines 75-78:

-          Objective should be rewrite, it is not clear.

-          Why have you used references in the objective?

-          You do not have to use references in the objective.

 Response: Thank you for your comment. We have rewritten the objective. We have deleted references in the objective.

(10)You should write a section with the design of the experiment at the beginning of “Material and methods”.

Response: Thank you for your comment. We have added the section with the design of the experiment.

(11)Line 86: water is liquid. Why do you use “g” for water instead of “L”?

Response: Thank you for your question. Because the density of the roast duck pickling solution is unknown, the dissolved extract is added to the pickling solution according to the mass concentration. In order to be consistent with the later pickling solution, we used “g” for water instead of “L”. We have modified it. Please check.

(12)Lines 82 and 85: you have to write the city and country of the companies.

Response: Thank you for your comment. We have added the city and country of the companies. Please check.

(13)Section “2.3.1. Total phenol content in natural extracts”: you should write the reference of the method.

Response: Thank you for your comment. We have modified the details and provided supplementary literature. Please check.

(14)Line 128: “et al” should be in italic.

Response: Thank you for your comment. We have modified the details. Please check.

(15)Lines 142-146:

-          Why have used these doses?

-          The doses are g/kg, kg of what? Because you have added the plant extract to the pickling liquid.

-          Why did you assess the effects on the formation of PAHs in duck skin? Do people eat the skin? In addition to this, aren’t there PAHs in duck?

-          You have to answer my questions and introduce the responses in that paragraph.

-          You have to rewrite that paragraph that is not clear.

Response: Thank you for your questions. We used these doses because many literatures had reported the antioxidant properties of RE, GTE, GSE and EBL extracts, but the recommended dosage of several extracts was very different, because these extracts were complex mixtures, the antioxidant effects of extracts from different sources were different in different foods, and the purity and effective components of extracts sold by different companies were also different.

Referring to relevant standards, it was found that the national food safety standard for the use of food additives (GB2760-2014) stipulates that the maximum use amount of RE in smoked, roasted, roasted meat and sauce meat products is 0.3 g/kg, and the use amount of tea polyphenols is 0.2 or 0.4 g/kg. There were no relevant standards for GSE and EBL as food additives. Therefore, the addition experiment was carried out with the addition amount of RE as the reference. In addition, because of the special production process of roast duck, it couldn’t be added in the process of chopping and mixing like minced meat products. Therefore, in combination with the process of industrialized production of roast duck, it was selected to add in the process of salting. Referring to the literature and the ingredient ratio of the "vegetable crispy" roast duck curing process in the cheap square(Beijing), considering that the extract may not be fully immersed in the sample, it is determined that a duck needs about 4 kg of pickling liquid for curing, and the plant extracts are made into the curing solution according to the proportion of 10 g, 30 g and 50 g per 4 kg of pickling liquid, which has the antioxidant effect by penetrating into the raw duck.

-          The doses are g/kg, kg of pickling liquid. The plant extracts were added to the roast duck marinade in proportion and penetrated into the raw duck to have an antioxidant effect.

-     According to previous studies, the existence of epidermis can effectively prevent PAHs from penetrating into biological tissues from flue gas during the roasting of roasted meat products. In particular, high molecular PAHs can be effectively intercepted by epidermis without being transferred into physical tissues. Only a small amount of naphthalene and acenaphthene with low molecular weight and low toxicity have been detected in roast duck meat, and no carcinogenic substances have been detected. Significantly different from duck skin samples. Therefore, in this study, duck meat will not be investigated.

(16)Section “2.4.1. Determination of the PAH content in roast duck”

-          You should write the reference of the method used.

-          Line 148: sample of what? Roast duck or duck skin?

-          You must indicate the meaning of the acronym MS the first time it appears in the text.

Response: Thank you for your comment. We have written the reference of the method used.

- Sample of duck skin, We have modified the details and provided Supplementary notes. 

-  We have indicated the meaning of the acronym MS in the text.

(17)Section “2.5. Data processing”:  you should explain the ANOVA and test to detect differences among samples.

 Response: Thank you for your comment. We have explained the ANOVA and test to detect differences among samples.

(18)Figures 1, 2, 3, 4, 5:

-          You should define the acronyms of the samples again, because the figures should be self-explanatory.

-          You should explain the meaning of different letters in the figure (significant differences?)

 Response: Thank you for your comment. We have defined the acronyms of the samples again and explained the meaning of different letters in the figures.

(19)Table 1:

-          You should define the acronyms of the samples again, because the tables should be self-explanatory.

-          You should explain the meaning of different letters in the table, significant differences among extracts or doses?

-          Indicate the units used.

Response: Thank you for your comment. We have defined the acronyms of the samples again. explain the meaning of different letters in the table, significant differences among doses?  μg/kg the units used.

Please check.

Figures 1, 2, 3, 4, 5:

(20)Lines 384-395: you should remove these lines because they are a summary of results and they are not conclusions.

Response: Thank you for your comment. We have removed these lines written again. Please check.

(21)Lines 401-402: the dose added is not small and could affect the flavor. In addition to this, you write “the flavor changes … can be investigated in future research”. Thus, you are incoherent and you are confirming that the flavor will change. You should rewrite these lines.

 Response: Thank you for your comment. We have written again. Please check.

(22)This section should be corrected according to the FOODS journal’s instructions. There are too many mistakes in the references.

Response: Thank you for your comment. We have read the FOODS journal’s instructions, and We have modified the details. Please check.

Reviewer 2 Report

  1.  
  2. Manuscript ID: foods-1772317
    Type of manuscript: Article
    Title: Effects of different natural plant extracts on the formation of PAHs
    in roast duck

 Dear Author,

The topic is interesting but the quality of the manuscript is poor. For that reason I have to suggest the rejection of this manuscript.  Some observations are as follows:

The material and methods section does not mention how many repetitions were carried out in the study. It is understood that this study was done with only with 2 kg of duck meat without repetitions.

The quality of figures is very poor. It is not possible to see the axis titles, treatments and comparison.

The figures title does not describe the treatments and the extract names in detail.

The significance is not clear. Do the literals mean the statistical comparison between concentrations or among extracts?

Writing is not very clear, per example:

Line 270-271 Sentence should be written again.

Line 384-390. Conclusions should be written as a general idea about the findings of the study making emphasis on the advantages and disadvantages of the results.

Line 398-400. Conclusions should not contain results of other studies.

 Author Response

Thank you for your comments on our manuscript, which are very helpful to improve our manuscript’s quality. We have made modification according to your suggestions. Please see the revised version uploaded for details. Below are the responses to comments.

(1) The material and methods section does not mention how many repetitions were carried out in the study. It is understood that this study was done with only with 2 kg of duck meat without repetitions.

 Response: Thank you for your comment. We have modified the details of “Material and methods”. Each extract was set with three concentration levels, and each level was set with three parallel levels, three repetitions for each parallel. The experimental materials were 2 kg raw duck embryos, and 13 roast ducks were actually used. Each experiment has three repetitions.

(2) The quality of figures is very poor. It is not possible to see the axis titles, treatments and comparison.

The figures title does not describe the treatments and the extract names in detail.

Response: Thank you for your comment. We have modified the details And added this information.

(3) The significance is not clear. Do the literals mean the statistical comparison between concentrations or among extracts?

Response: Thank you for your comment. The literals mean the statistical comparison among extracts. We have modified the details. Please check.

(4)Writing is not very clear, per example:

Line 270-271 Sentence should be written again.

 Response: Thank you for your comment. We have written again.

(5)Line 384-390. Conclusions should be written as a general idea about the findings of the study making emphasis on the advantages and disadvantages of the results.

Response: Thank you for your comment. We have written again.

(6) Line 398-400. Conclusions should not contain results of other studies.

Response: Thank you for your comment. We have modified it. Please check.

Reviewer 3 Report

Foods

foods-1772317

Effects of different natural plant extracts on the formation of PAHs in roast duck

Dear Editor,

The article deals with the determination of the effects of different natural plant extracts on the formation of PAHs in roast duck. The topic is good. The paper has been well designed and written. It can be accepted after necessary corrections are done. My questions and comments are below;

-       The formation mechanisms of PAHs should be discussed in more detail.

-       Line 63: “heterocyclic amines” should be “heterocyclic aromatic amines”

-       Line 69: “heating” should be “cooking”

-       Mention the aim of the study clearly in the last part of the introduction section.

-       Line 86: All of these extracts were water-soluble?

-       Line 235: What are the possible reasons for the recovery rate to exceed 100%?

-       Line 236: Give the reference!

-       Have the researchers made correlations?

-       Are there any other mechanisms that prevent/reduce the occurrence of PAHs? If any, these mechanisms should be mentioned in the text.

Author Response

Dear Reviewer:

Thank you for your comments on our manuscript, which are very helpful to improve our manuscript’s quality. We have made modification according to your suggestions. Please see the revised version uploaded for details. Below are the responses to comments.

(1)        The formation mechanisms of PAHs should be discussed in more detail.

Response: Thank you for your comment. We have modified the details and provided supplementary literature. Please check.

(2)       Line 63: “heterocyclic amines” should be “heterocyclic aromatic amines”

Response: Thank you for pointing out this error. We have corrected it. Please check.

(3)       Line 69: “heating” should be “cooking”

Response: Thank you for pointing out this error. We have corrected it. Please check.

 ï¼ˆ4)     Mention the aim of the study clearly in the last part of the introduction section.

Response: Thank you for your comment. We have added the details of of the “introduction” . Please check.

(5)     Line 86: All of these extracts were water-soluble?

Response: Thank you for your comment. All of these extracts were water-soluble We have added the details of “Material and methods”. Please check.

(6)         Line 235: What are the possible reasons for the recovery rate to exceed 100%?

Response: Thank you for your question. It may be caused by matrix effect (me), that is, in the process of sample detection, due to the presence of matrix residual components other than the target to be tested, the phenomenon of direct or indirect impact on the target is represented by ion enhancement. When the pollution of polycyclic aromatic hydrocarbons in roast duck products is determined by GC-MS, the endogenous substances (lipids, pigments, sugars, etc.) in the samples will affect the response of the target substances. Although the effective pretreatment, extraction and purification technology is one of the measures to reduce the matrix effect, it still can not completely eliminate the matrix effect.

(7)        Line 236: Give the reference!

Response: Thank you for your comment. We have provided reference.

(8)     Have the researchers made correlations?

Response: Thank you for your question. We haven't carried out correlation experiments yet, because we haven't analyzed and verified the active ingredients with antioxidant effect in the four extracts, and the concentration of the active ingredients in the purchased extracts has not been accurately quantified. We just simply compared and studied the effects of three different levels of addition on the inhibition rate, so we haven't carried out correlation analysis, but the correlation can be used as the next research content.

(9)      Are there any other mechanisms that prevent/reduce the occurrence of PAHs? If any, these mechanisms should be mentioned in the text.

Response: Thank you for your comment. We have modified the details and provided supplementary literature. For example, during the barbecue process of meat products, the processing temperature and time should be reasonably controlled; barbecue mode was changed and raw meat with low fat content can also effectively reduced the content of PAHs in barbecue meat products.

Round 2

Reviewer 1 Report

Most of my comments have been addressed. However, some points have not been carried out.

In order to make the paper published in a journal with the quality of FOODS, the following issues should be corrected. 

  ABSTRACT

You should write a clear objective in the abstract. You have not done it.

 MATERIAL AND METHODS

What test did you perform to indicate the differences among samples? You should indicate.

 RESULTS AND DISCUSSION

The resolution of the figures is bad. You should improve the quality of figures.

  REFERENCES

This section should be corrected according to the FOODS journal’s instructions. There are too many mistakes in the references. For instance, the name of Journals should be abbreviated.

Author Response

Dear Reviewer:

Thank you for your comments on our manuscript, which are very helpful to improve our manuscript’s quality. We have made modification according to your suggestions. Please see the revised version uploaded for details. Below are the responses to comments.

(1) ABSTRACT

You should write a clear objective in the abstract. You have not done it.

Response: Thank you for your comment. We have added content in the abstract to make the objective more clear (line 11-line 14).

(2) MATERIAL AND METHODS

What test did you perform to indicate the differences among samples? You should indicate.

 Response: Thank you for your comment. The results were statistically analyzed by ANOVA (p<0.05). Comparison of the mean values was performed using Duncan’s test. We have added the test to indicate the differences among samples (line 214-line 218), please check it.

(3) ESULTS AND DISCUSSION

The resolution of the figures is bad. You should improve the quality of figures.

Response: Thank you for your comment. We have redrawn the figures and improved the resolution.

Reviewer 2 Report

Dear Authors,

Thank you for making all the suggested corrections.

The manuscript has improved significantly after modifications.

No further changes are needed from my point of view.

Author Response

Dear Reviewer:

Thank you for your recognition of this article. Wish you every success in your work.
